

# MBAN: multi-branch attention network for small object detection

Li Li, Shuaikun Gao, Fangfang Wu and Xin An

School of Information and Electrical Engineering, Hebei University of Engineering, Handan, Hebei, China

## ABSTRACT

Recent years small object detection has seen remarkable advancement. However, small objects are difficult to accurately detect in complex scenes due to their low resolution. The downsampling operation inevitably leads to the loss of information for small objects. In order to solve these issues, this article proposes a novel Multi-branch Attention Network (MBAN) to improve the detection performance of small objects. Firstly, an innovative Multi-branch Attention Module (MBAM) is proposed, which consists of two parts, *i.e.* Multi-branch structure consisting of convolution and maxpooling, and the parameter-free SimAM attention mechanism. By combining these two parts, the number of network parameters is reduced, the information loss of small objects is reduced, and the representation of small object features is enhanced. Furthermore, to systematically solve the problem of small object localization, a pre-processing method called Adaptive Clustering Relocation (ACR) is proposed. To validate our network, we conducted extensive experiments on two benchmark datasets, *i.e.* NWPU VHR-10 and PASCAL VOC. The findings from the experiment demonstrates the significant performance gains of MBAN over most existing algorithms, the mAP of MBAN achieved 96.55% and 84.96% on NWPU VHR-10 and PASCAL VOC datasets, respectively, which proves that MBAN has significant performance in small object detection.

## INTRODUCTION

Object detection remains one of the most challenging topics, maintaining a large number of computer vision tasks, for instance, automatic driving (*Dai, 2019*), face recognition (*Yu et al., 2022*), defect detection (*Jing et al., 2020*), remote sensing image detection (*Xu & Wu, 2020*). However, small object detection remains challenging because of limited information and its sensitivity to background interference, which hinders its further development in real-world scenarios.

At present, there are mainly two types of object detection algorithms based on deep learning, namely two-stage algorithms R-CNN (*Girshick et al., 2014*), SPP-Net (*He et al., 2015*), Fast R-CNN (*Girshick, 2015*), Faster R-CNN (*Ren et al., 2015*), R-FCN (*Dai et al., 2016*) and Mask R-CNN (*He et al., 2017*), one-stage algorithms YOLO (*Redmon et al., 2016*), SSD (*Liu et al., 2016*), RetinaNet (*Lin et al., 2017*) and EfficientDet (*Tan, Pang & Le, 2020*). The two-stage object detection algorithms first use a region proposed module to offer all regions, and then select the most matched one as the object. Although these

Corresponding author
Xin An, anxin@hebeu.edu.cn

algorithms achieve better detection accuracy, they suffer from slow detection speed and inadequate real-time performance. The one-stage algorithms use an end-to-end convolutional neural network to detect objects. One-stage object detection algorithms not only maintain high detection accuracy but also have the advantages of fast detection speed and strong real-time performance. Therefore, we select a one-stage framework, *i.e.* YOLOv5l (*Cui et al., 2023*), as a baseline in the study. However, YOLOv5l as a generalized object detection network model does not show significant advantages in identifying small objects. The dimensions of small objects are relatively small, while YOLOv5l has a large downsampling factor, which can lead to information loss of small objects during the downsampling process, making it difficult to learn the feature information of small objects through deeper feature maps. Therefore, it is crucial to improve the ability to better accommodate scenarios that involve small objects.

It is a natural thought to aggregate low-level and high-level features to enhance the localization information of high-level features. Meanwhile, in order to decrease transmission loss of low-level feature information, a "short-cut" has been added between the lowest and highest levels, *e.g.* PANet (*Liu et al., 2018*). Moreover, downsampling entails the loss of valuable information for small objects. This issue is particularly pronounced for objects with dimensions smaller than $32 \times 32$ pixels, as they can easily blend into the background, thereby impeding small object detection (*Yan et al., 2022*).

With the development of technology, the demand for detecting small objects is constantly increasing. In production, many places require the detection of small objects, such as detecting defects in product components and detecting pollutants in the air. Due to the low resolution of small objects, they may appear blurry and have unclear details in the image, making it difficult to accurately recognize the features of small objects. Meanwhile, small objects may be affected by the environment, which may reduce their visibility and make them more difficult to detect.

To resolve the aforementioned problems, this article proposes an MBAN. Our method mitigates the loss of small object information when the downsampling process by incorporating a carefully designed MBAM. Additionally, it leverages the ACR to localize the position of small objects, notably enhancing the accuracy of small object detection. We performed a wide range of experiments on the NWPU VHR-10 and PASCAL VOC datasets to verify the excellent performance of the MBAN compared to mainstream algorithms such as SSD, EfficientNet-YOLOv3, YOLOv3, YOLOv4, GhostNet-YOLOv4, and YOLOv5l.

The main contributions of this work can be summarized as follows:

(1) A novel MBAN is proposed for small object detection, where the core MBAM of MBAN reduces the problem of small object information loss during downsampling, and the network significantly improves the detection accuracy of small objects.

(2) The use of ACR method to accurately locate the position of small objects significantly improves the accuracy of small object detection.

(3) The proposed MBAN conducted experiments on NWPU VHR-10 and PASCAL VOC datasets. The experimental results indicate that the MBAN significantly improves detection accuracy in small object detection without sacrificing detection speed.

The subsequent sections of this article are structured as outlined below. "Related Work" describes a review of the literature. "Proposed Method" outlines unique MBAN. "Experiments and Discussion" showcases the experimental results and discussion. "Qualitative Analysis" displays visualization results. Last, "Conclusion" summarizes in this article.

## RELATED WORK

Within this part, we review the related work on small object detection methods, attention mechanism, and multi-branch attention methods.

### Small object detection methods

Small object detection has always been a challenge and a focal point in computer vision. Driven by deep learning, significant breakthroughs have been made in this area. *Zhang et al. (2021)* argued that the problem of small objects and object aggregation leads to less extractable information. Most studies mainly utilize large networks to enhance the detection accuracy, which results in the problem of large model size and sluggish detection. Therefore, a combination based on MobileNet v2 and depthwise separable convolution was introduced to decrease the quantity of model size. Furthermore, the semantically and scale inconsistent features are fused using an improved attentional feature fusion (AFFM) module, which aims to enhance the model's accuracy for small objects. *Gong et al. (2021)* found that the detection of small objects is impacted by the top-down connections between neighboring layers of the FPN, therefore, they proposed a fusion factor to control the information passed from the deeper layers to the shallower layers, significant performance improvement was achieved on small object detection datasets, for instance, TinyPerson and TinyCityPersons. *Yang, Huang & Wang (2022)* believed that the way to facilitate small object detection was to leverage high-resolution images or feature maps. Nevertheless, such an approach entails computationally intensive processes. In response to this, they introduced QueryDet, which quickens feature-pyramid based object detectors inference speed through the use of a revolutionary query technique. Significant enhancement in detection performance was observed on the baseline datasets COCO and VisDrone. *Zhang & Shen (2022)* offered a multi-stage feature enhancement pyramid network to effectively resolve the problem of small-scale objects blurring and large-scale objects changing detected in remote sensing images. The network solves the issue of feature map fusion of neighboring stages by using Feature Enhancement Module (FEM) and Content-Aware Feature Up-Sampling (CAFUS).

Although these methods enhance the detection performance of small objects, persistent challenges in the realm of small object detection necessitate further in-depth research. Firstly, small objects have small dimensions, limited information, and extracting discriminative features is difficult. Secondly, small objects have high requirements for positioning accuracy, and downsampling can easily lead to information loss and prevent accurate positioning of small objects. Thirdly, small objects are prone to interference from background information, and the changes in light and the aggregation of small objects in complex environments make it difficult to accurately detect them.

## Attention mechanism

Attention Mechanism in Deep Learning (Attention Mechanism (*Chen et al., 2021b*)) is a method that mimics the visual and cognitive system of humans, which enables neural networks to closely concentrate on relevant parts when processing input data. The neural networks learn automatically and selectively concentrate on the significant information to promote the performance and generalization of the network by introducing attention mechanism. Currently, common attention mechanisms include SE, CBAM, ECA, *etc.*

*Hu, Shen & Sun (2018)* introduced the Squeeze-and-Excitation (SE) block adaptively recalibrate channel feature responses. This block improves detection performance. *Woo et al. (2018)* devised Convolutional Block Attention Module (CBAM) deduces the attention map along the channel and spatial order before multiplying it by the input feature map to enable adaptive feature refining. *Wang et al. (2020)* devised an Efficient Channel Attention (ECA) module, this module preserves performance while avoiding dimensionality reduction and efficiently capturing cross channel interaction information.

Although adding the above attention mechanisms to the algorithm can improve detection performance, they all increase the quantity of algorithm parameters. Compared with the aforementioned attention mechanisms, the SimAM attention mechanism (*Yang et al., 2021*) does not require additional learning parameters, therefore, this article adopts the SimAM parameter-free attention mechanism, which not only improves the detection performance of the algorithm, but also does not increase the quantity of algorithm parameters.

## Multi-branch attention methods

Multi-branch attention method is an approach that utilizes multiple attention branches to focus on different aspects of input, thereby capturing richer feature information and improving network detection capabilities. *Chen et al. (2019)* designed a multi-branch attentional neural network for separate detection of vehicles and license plates using different convolutional layers, thereby eliminating mutual interference between each other. By incorporating a task-specific anchor design strategy, as well as attention mechanisms and feature-fusion strategies, the network improves the detection accuracy of targets. *Chen et al. (2021a)* proposed a Multi-Branch Local Attention Network (MBLANet). It combines the Convolutional Local Attention Module (CLAM) with the ResNet50 deep residual network. Additionally, it parallelly integrates the Convolutional Channel Attention Module (CCAM) and the Local Spatial Attention Module (LSAM) to form CLAM. This combination facilitates better extraction of crucial feature information. To address the challenges posed by densely arranged and arbitrarily oriented in remote sensing images. *Zhou, Wang & Chen (2023)* introduced a Single-Stage Rotation Object Detector named FTANet, which primarily employs Balanced Feature Pyramid (BFP) and Triple Branch Attention (TBA) modules. BFP integrates information from different levels to mitigate the impact of complex backgrounds on targets in remote sensing images. TBA enhances classification accuracy and prediction of rotated anchors angles by decoupling the angle parameter from five parameters and using the attention mechanism to highlight key

foreground features. It enriches the semantic information of images and highlights key features, resulting in higher detection accuracy.

Although the aforementioned multi-branch attention methods improve the performance of detection of small objects, they also increase the number of network parameters. Therefore, we propose a novel MBAN to enhance the detection capability of small objects, which not only does not increase the number of the network parameters, but also improves the performance of the network detection of small objects.

## PROPOSED METHOD

Within this part, we first introduce MBAN. Subsequently, we elaborate on the core component of MBAN: MBAM. Finally, we further introduce pre-processing method ACR to methodically address the object location issue.

### The multi-branch attention network

The overall structure of the presented MBAN comprises three components: backbone feature extraction network (Backbone), feature pyramid reinforcement network (Neck), and prediction network (Prediction). The Neck part employs MBAM for downsampling operations. MBAM consists of Multi-branch structure and SimAM attention mechanism, which enables the network to prioritize salient information concerning small objects, thereby enhancing the representation of their features and improving detection accuracy. The network architecture of MBAN as shown in Fig. 1. The structure of MBAN is shown in Table 1.

Backbone consists of the Input, Focus structure, CBS, SPP structure and CSP structure. In the Backbone phase, the network acquires three feature layers named feat1, feat2, and feat3. These layers have the following shapes: feat1 = (80, 80, 256), feat2 = (40, 40, 512) and feat3 = (20,20,1,024). These feature layers serve as crucial inputs for constructing the subsequent stages of the network.

Neck in YOLOv5l acts as a robust feature extraction network that undergoes feature fusion on three key feature layers obtained from the backbone. This process aims to combine feature information from different scales effectively. The architecture of PANet is still used in YOLOv5l, where the features are not only upsampling for feature fusion, but also downsampling again for feature fusion.

In this article, we use the detection of NWPU VHR-10 dataset as a case study. We start with a $640 \times 640$ input image, which undergoes a series of operations including a Backbone network, a Neck, and subsequent feature integration and channel adjustment through Prediction. These operations yield three sets of feature layers, each characterized by dimensions (80, 80, 45), (40, 40, 45), and (20, 20, 45). Here, the number 45 can be divided into $3 \times (10 + 1 + 4)$, 3 is the existence of three priori boxes for YOLOv5l at each feature point of the feature layer, 10 represents the NWPU VHR-10 dataset is categorized into 10 distinct categories, 1 indicates whether the box contains objects or not, and 4 represents the tuning parameters of the predicted box: centroid coordinates x, y and the width and height of the box w, h. The category confidence scores for each grid can be formulated as Eq. (1).

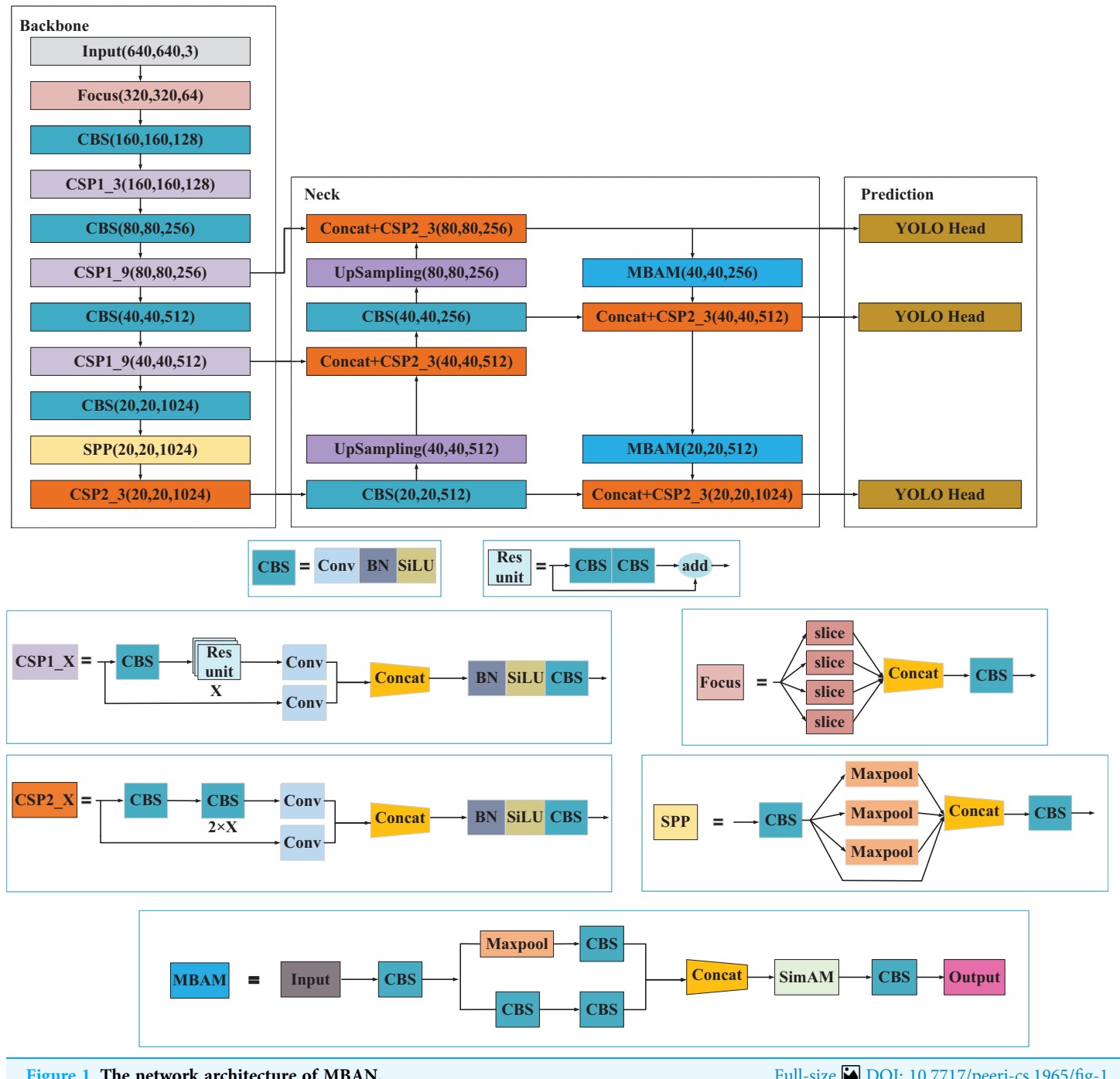

**Figure 1** The network architecture of MBAN.

$$P_r(scores) = P_r(Class_i|object) * P_r(object) * IoU_{pred}^{truth} \tag{1}$$

where $P_r(Class_i|object)$ denotes the probability that the grid corresponds to $i$ category. $P_r(object)$ denotes the probability of the object's existence. $P_r(object) = 1$ if there is an

**Peer**J Computer Science

**Table 1 The structure of MBAN.**

| Backbone | Neck | | Prediction |
|---|---|---|---|
| Input (640,640,3) | – | – | – |
| Focus (320,320,64) | – | – | – |
| CBS (160,160,128) | – | – | – |
| CSP1_3 (160,160,128) | – | – | – |
| CBS (80,80,256) | CSP2_3 (80, 80, 256) | – | CBS (80, 80, 45) |
| CSP1_9 (80,80,256) | UpSampling (80, 80, 256) | MBAM (40, 40, 256) | – |
| CBS (40,40,512) | CBS (40, 40, 256) | CSP2_3 (40, 40, 512) | CBS (40, 40, 45) |
| CSP1_9 (40,40,512) | CSP2_3 (40, 40, 512) | – | – |
| CBS (20,20,1,024) | – | – | – |
| SPP (20,20,1,024) | UpSampling (40, 40, 512) | MBAM (20, 20, 512) | – |
| CSP2_3 (20,20,1,024) | CBS (20, 20, 512) | CSP2_3 (20, 20, 1,024) | CBS (20, 20, 45) |

object center falling at that grid point, and 0 otherwise. $IoU_{pred}^{truth}$ is the intersection over union of the predicted box and the real box.

The loss function includes of three sections: object confidence loss, object classification loss and object localization loss. The loss function is shown in Eq. (2).

$$L_{all} = \lambda_{conf}L_{conf} + \lambda_{cls}L_{cls} + \lambda_{loc}L_{loc} \tag{2}$$

where $L_{all}$ contains three hyperparameters $\lambda_{conf}$, $\lambda_{cls}$ and $\lambda_{loc}$, which represent the weights of the three parts, and the corresponding weights are 1.0, 0.5 and 0.05, respectively.

The Intersection over Union (IoU) indicates evaluation of the extent of overlap between the predicted and real detection boxes in object detection. IoU is calculated using Eq. (3). Generally, as the IoU approaches 1, it means that the detection results are more accurate. Since the non-sensitivity of IoU to the scale and overlap rate of the objects. GIoU loss (*Rezatofighi et al., 2019*) is used as the loss function of the bounding box in YOLOv5l. The computation process of GIoU is shown in Eq. (4), and the computation formula of the loss of object localization is shown in Eq. (5).

$$IoU = \frac{|A \cap B|}{|A \cup B|} \tag{3}$$

where $A$ and $B$ represent the predicted box and the real box, respectively.

$$GIoU = IoU - \frac{|C - (A \cup B)|}{|C|} \tag{4}$$

$$L_{loc} = GIoU_{Loss} = 1 - GIoU = 1 - IoU + \frac{|C - (A \cup B)|}{|C|} \tag{5}$$

where $C$ is the smallest bounding box that covers both the predicted box and the real box.

The confidence loss for the objects employs the BCE (Binary Cross Entropy) loss as illustrated in Eq. (6).

$$L_{conf} = -\sum_{i=0}^{S*S}\sum_{j=0}^{B} I_{ij}^{obj}[\widehat{C_i^j}log(C_i^j) + (1 - \widehat{C_i^j})log(1 - C_i^j)] -$$

$$\lambda_{nobj}\sum_{i=0}^{S*S}\sum_{j=0}^{B} I_{ij}^{nobj}[\widehat{C_i^j}log(C_i^j) + (1 - \widehat{C_i^j})log(1 - C_i^j))] \qquad (6)$$

where $S * S$ can take three different values depending on the image size, for instance, with an input image size of $640 \times 640$, they are $20 \times 20$, $40 \times 40$, and $80 \times 80$, which illustrates the quantity of grids on the feature maps produced by YOLOv5l at three different scales. $B$ is the quantity of priori boxes generated for each grid. $I_{ij}^{obj}$ specifies whether the *jth* priori box of the *ith* grid has an object. If the condition is satisfied, $I_{ij}^{obj}$ is 1; otherwise it is 0. $I_{ij}^{nobj}$ specifies whether the *jth* priori box of the *ith* grid does not contain the objects. If it does not contain, $I_{ij}^{nobj}$ is 1; otherwise it is 0. $\widehat{C_i^j}$ and $C_i^j$ denote the real box confidence and the predicted box confidence, respectively. $\lambda_{nobj}$ denotes the weight coefficient that does not include the object confidence.

The object classification loss is shown in Eq. (7).

$$L_{cls} = -\sum_{i=0}^{S*S} I_{ij}^{obj}\sum_{c\in classes} [\hat{P_i^j}log(P_i^j) + (1 - \hat{P_i^j})log(1 - P_i^j)] \qquad (7)$$

where $S * S$ and $I_{ij}^{obj}$ are consistent with Eq. (6), $c$ represents the object category, and $\widehat{P_i^j}$ and $P_i^j$ denote the probability that the object in the *jth* predicted box within the *ith* grid corresponds to the real value and the predicted value of a specific category, respectively.

## Multi-branch attention module

We further propose an MBAM to better extract small object features while decreasing the quantity of network parameters and raising the detection accuracy of the network. To be specific, the execution of this module proceeds as outlined below. Firstly, channel dimension and parameter count of the feature map are reduced using $1 \times 1$ convolution. Secondly, the module employs two branches: the first branch reduces the input feature map size by half through max-pooling with a stride of 2 and a kernel size of 2, focusing on learning edge information; the second branch resizes the input feature map to half its size using a $3 \times 3$ convolution with a stride of two. Both branches utilize a $1 \times 1$ convolution to learn small object feature information and facilitate cross-channel interaction and integration. Thirdly, the outputs from the two branches are fused to obtain feature maps with rich semantic information, enhancing feature expression in small objects. To resist confusing information and focus on the key feature information of interest, we apply the SimAM parameter-free attention mechanism. Finally, channel dimension is adjusted to match the input channel dimension using a $1 \times 1$ convolution to obtain the ultimate feature map. Figure 2 presents the structure of the MBAM.

Specially, the parameter-free SimAM attention mechanism enhances the algorithmic focus on the significant information of the objects. The SimAM attention mechanism can balance the feature weights more comprehensively and efficiently, and reduce the
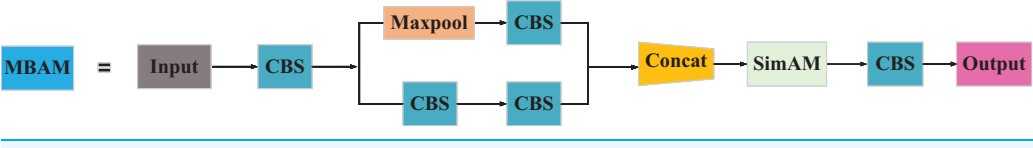

**Figure 2 The structure of MBAM.**

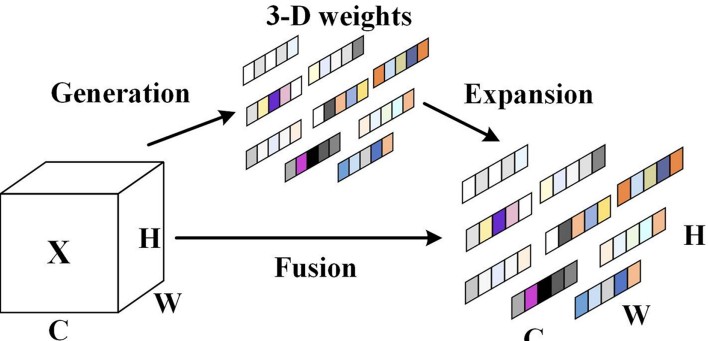

**Figure 3 The SimAM attention mechanism architecture.**

background feature weights. Figure 3 illustrates the architecture of the SimAM attention mechanism.

The SimAM attention mechanism is capable of assigning higher weights to important neurons as a way of presenting the importance of these important neurons, and the *sigmoid* is added to limit excessively large values in $e_t^*$. The computational procedure of the SimAM attention mechanism is shown in Eq. (8).

$$\tilde{X} = sigmoid\left(\frac{1}{E}\right) \odot X \tag{8}$$

where $X$ is the input feature, $\tilde{X}$ is the output feature, $e_t^*$ is the energy function, and $E$ groups all $e_t^*$ across channel and spatial dimensions.

The SimAM attention mechanism identifies more important neurons by measuring the linear separability between different neurons, and defines the energy function of each neuron as shown in Eq. (9).

$$e_t(\omega_t, b_t, y, x_i) = (y_t - \hat{t})^2 + \frac{1}{M-1}\sum_{i=1}^{M-1}(y_o - \widehat{x_i})^2 \tag{9}$$

where $\hat{t} = \omega_t t + b_t$ and $\widehat{x_i} = \omega_t x_i + b_t$ are linear transforms of $t$ and $x_i$, $t$ and $x_i$ are the object neuron and other neurons within the individual channel of the input feature $X \in \mathbb{R}^{C \times H \times W}$, $i$ is index in the spatial dimension. $M = H \times W$ signifies the number of neurons on that channel. $\omega_t$ and $b_t$ are weights and biases, respectively. $y_o$ and $y_t$ are two different values, for simplicity, set $y_t$ and $y_o$ to 1 and −1 respectively, after regularization of the energy function in Eq. (9). The final energy function is shown in Eq. (10).

$$e_t(\omega_t, b_t, y, x_i) = \frac{1}{M-1} \sum_{i=1}^{M-1} (-1 - (\omega_t x_i + b_t))^2 + (1 - (\omega_t t + b_t))^2 + \lambda \omega_t^2 \qquad (10)$$

Equation (10) has a fast closed-form solution with respect to $\omega_t$ and $b_t$, which can be obtained by the following Eqs. (11) and (12).

$$\omega_t = -\frac{2(t - \mu_t)}{(t - \mu_t)^2 + 2\sigma_t^2 + 2\lambda} \qquad (11)$$

$$b_t = -\frac{1}{2}(t + \mu_t)\omega_t \qquad (12)$$

where $\mu_t = \frac{1}{M-1}\sum_{i=1}^{M-1} x_i$ and $\sigma_t^2 = \frac{1}{M-1}\sum_i^{M-1}(x_i - \mu_t)^2$ are mean and variance calculated over all neurons except $t$ in that channel, respectively.

In order to avoid iteratively calculating $\mu$ and $\sigma$ for each position, the computation of $e_t^*$ is represented by Eq. (13), as the energy $e_t^*$ decreases, the difference between neuron $t$ and surrounding neurons increases, thereby indicating the greater significance of neurons, so the significance of each neuron can be derived by $1/e_t^*$.

$$e_t^* = \frac{4(\hat{\sigma}^2 + \lambda)}{(t - \hat{\mu})^2 + 2\hat{\sigma}^2 + 2\lambda} \qquad (13)$$

where $\lambda$ is the hyperparameter, $\hat{\mu} = \frac{1}{M}\sum_{i=1}^{M} x_i$ and $\hat{\sigma}^2 = \frac{1}{M}\sum_{i=1}^{M}(x_i - \hat{\mu})^2$. After manipulating the neurons, the neurons with more critical information are given more important weights, which enhances the network's detection accuracy without introducing additional parameters.

### Adaptive clustering relocation

The YOLOv5l object detection algorithm originally utilized prior boxes derived from clustering the COCO dataset. However, this approach proved inadequate for detecting small objects due to their diminutive size. Consequently, it is necessary to cluster their respective appropriate prior boxes according to the characteristics of different datasets. To resolve this problem, this article introduces the K-means clustering algorithm (*Yuan & Yang, 2019*), which takes the distance as a measure, furthermore, as the distance between data decreases, the similarity increases. The final result is obtained by constantly updating the position of the clustering center. The K-means clustering algorithm encompasses the following steps:

(1) Randomly choose one sample as the original clustering center.

(2) Divide each sample dataset into the nearest cluster centers.

(3) The cluster center to which each sample belongs is updated.

(4) Repeat steps (2) and (3), until convergence, and output the results of the clustering algorithm.

The priori boxes of COCO dataset are (10, 13), (16, 30), (33, 23), (30, 61), (62, 45), (59, 119), (116, 90), (156, 198), (373, 326) in order from smallest to largest. In this article, nine prior boxes are clustered for each dataset using K-means clustering algorithm, and three

**Table 2** The clustering results on the NWPU VHR-10 dataset.

| Detection layer | Prior box size | | |
|---|---|---|---|
| 20 × 20 | (56, 84) | (89, 113) | (170, 297) |
| 40 × 40 | (32, 53) | (49, 56) | (35, 82) |
| 80 × 80 | (20, 32) | (27, 42) | (43, 37) |

**Table 3** The clustering results on the PASCAL VOC dataset.

| Detection layer | Prior box size | | |
|---|---|---|---|
| 20 × 20 | (235, 440) | (436, 296) | (528, 547) |
| 40 × 40 | (92, 85) | (107, 274) | (194, 179) |
| 80 × 80 | (40, 34) | (23, 64) | (49, 144) |

prior boxes are distributed to every detection layer, and the prior boxes of the NWPU VHR-10 dataset after clustering are illustrated in Table 2.

The priori boxes of the PASCAL VOC dataset after clustering are displayed in Table 3.

## EXPERIMENTS AND DISCUSSION

In the remaining part, we conducted experiments using MBAN on the PASCAL VOC and NWPU VHR-10 datasets and compared MBAN with some advanced algorithms. Additionally, we devise ablation experiments to investigate the performance of the methods introduced in MBAN.

### Implementation details

The experimental environment is set up as shown below: the processor is 13th Gen Intel Core i5-13500H, the graphics card is RTX 4050, 16 GB RAM, and the operating system is Windows 11. The experiments are conducted using Pycharm compilation software, Python programming language, and the deep learning framework is Pytorch. The experiments are conducted using the Adam optimization ware (*Yi, Ahn & Ji, 2020*) to optimize the network, and the dimensions of the input image for training all algorithms are set to (640, 640). The experimental hyperparameters of this article are shown in Table 4.

### Datasets

The PASCAL VOC dataset (*Everingham et al., 2010*) is an open object detection dataset that includes 20 object categories of different scales and poses. In this article, we use the trainval set of PASCAL VOC 2007 (*Everingham et al., 2007*) and PASCAL VOC 2012 (*Everingham et al., 2012*) (16,551 images) to train the MBAN. The test set of PASCAL VOC 2007 (4,952 images) is used to evaluate the performance.

The NWPU VHR-10 dataset (*Cheng & Han, 2016; Cheng et al., 2014; Cheng, Zhou & Han, 2016*) is an open remote sensing dataset consisting of 800 images, it contains 10

**Table 4 The experimental hyperparameters.**

| Experimental hyperparameters | Value |
|---|---|
| Initial learning rate | 0.001 |
| Minimum learning rate | 0.00001 |
| Freezing stage epochs | 50 |
| Unfreezing stage epochs | 50 |
| Total stage epochs | 100 |
| Freezing stage batch size | 4 |
| Unfreezing stage batch size | 2 |
| Optimizer type | Adam |
| Momentum | 0.937 |

object categories. In the training phase, the NUPU VHR-10 dataset is divided into trainval set and test set in the ratio of 9:1, and 90% in the trainval set is used for train and 10% for validation.

## Evaluation metrics

In this article, frame per second (FPS), average precision (AP), mean average precision (mAP), $F1$ score, $P$ and $R$ are used as the evaluation metrics to evaluate the algorithm performance. FPS is the quantity of frames per second of processed images. AP represents the area surrounded by a P-R curve and coordinate axes. The P-R curve is the curve plotted with the recall as the horizontal axis and the precision as the vertical axis. Precision and Recall are defined using Eqs. (14) and (15).

$$Precision = \frac{TP}{TP + FP} \tag{14}$$

$$Recall = \frac{TP}{TP + FN} \tag{15}$$

where $TP$ denotes the quantity of accurately detected positive samples, $FP$ represents the amount of inaccurately detected positive samples, and $FN$ represents the quantity of inaccurately detected negative samples, *i.e.* the quantity of missed detections.

$$AP = \int_0^1 P(R)dR \tag{16}$$

where $AP$ represents the area surrounded by a P-R curve and coordinate axes. It can be calculated by Eq. (16).

The mAP is the average value of different kinds of AP. The mAP can be calculated by Eq. (17).

$$mAP = \frac{AP_1 + AP_2 + \cdots + AP_n}{n} \tag{17}$$

where $n$ is the amount of all object categories.

$$F1 = 2 \times \frac{R \times P}{R + P} \tag{18}$$

where $F1$ is the reconciled mean of $P$ and $R$, it can be calculated by Eq. (18).

# RESULTS AND DISCUSSION

## Discussion of results on NWPU VHR-10 dataset

To showcase the efficacy of the MBAN, we compared the experimental results of some mainstream algorithms on the NWPU VHR-10 dataset, it includes YOLOv3, YOLOv4, SSD, EfficientNet-YOLOv3, GhostNet-YOLOv4, and YOLOv5l. Table 5 presents the comparison results. The best results are highlighted in bold.

From Table 5, it is evident that the mAP of MBAN achieves 96.55%, and the detection speed remains at 28 FPS. In comparison to YOLOv5l, the network size has decreased by 4.05 MB and the number of parameters has been reduced by 2.3%, MBAN's mAP has improved by 2.16%, while recall (R), precision (P), and F1 have seen improvements of 0.69%, 1.29%, and 0.01%, respectively, fully demonstrating the efficacy of this network. In contrast with SSD, although the network size increased by 75.19 MB and the number of parameters increased by 74.97%, MBAN's mAP improved by 6.26%, and detection speed increased by 14 FPS. Compared to YOLOv3, MBAN's mAP increased by 5.35%, while the network size decreased by 60.86 MB, the number of network parameters reduced by 25.75%, and the detection speed improved by 4 FPS. Despite the increase in network size by 134.35 MB and the increase in the number of parameters by 38.78 M compared to EfficientNet-YOLOv3, with a decrease in detection speed by 26 FPS, MBAN's mAP increased by 7.05%. Compared to YOLOv4, MBAN improved mAP by 4.01%, reduced network size by 70.07 MB, reduced network parameter count by 28.53%, and improved detection speed by 9 FPS. In comparison with GhostNet-YOLOv4, despite the increase in network size by 131.86 MB and the increase in the number of parameters by 34.57 M, with a decrease in detection speed by 33 FPS, MBAN's mAP increased by 7.83%, still meeting real-time detection requirements.

This article compares the detection accuracy of MBAN and some mainstream algorithms for each category on the NWPU VHR-10 dataset, as shown in Table 6. Out of the seven algorithms mentioned, bold indicates the highest AP for a single object category. It is evident that MBAN outperforms the mainstream algorithms in the majority of categories, indicating its superior accuracy. Compared to SSD, MBAN has improved detection accuracy in eight categories. Compared with YOLOv3, although MBAN has improved detection accuracy in only six categories, its network structure is complex and has a large number of parameters. Compared with EfficientNet-YOLOv3, MBAN has improved detection accuracy in nine categories and the detection accuracy is equal in one category. Compared with YOLOv4, MBAN detection accuracy has only decreased in one category, while achieving excellent detection results in other categories. Compared with GhostNet-YOLOv4, MBAN has the same detection accuracy in two categories and improved detection accuracy in eight categories. In contrast to the YOLOv5l, the detection accuracy is equal in four categories and improved in three categories. MBAN's detection accuracy for small object categories of ship and vehicle has increased by 1.22% and 0.57%,

**Table 5  The experimental results of different algorithms on the NWPU VHR-10 dataset.**

| Algorithm | R/% | P/% | F1 | mAP/% | Size/MB | Params/M | FPS |
|---|---|---|---|---|---|---|---|
| SSD | 82.90 | 89.35 | 0.87 | 90.29 | 100.27 | 26.29 | 14 |
| YOLOv3 | 89.60 | **91.53** | 0.90 | 91.20 | 236.32 | 61.95 | 24 |
| EfficientNet-YOLOv3 | 77.95 | 86.57 | 0.80 | 89.50 | **41.11** | **7.22** | 54 |
| YOLOv4 | 89.86 | 90.92 | 0.90 | 92.54 | 245.53 | 64.36 | 19 |
| GhostNet-YOLOv4 | 84.22 | 86.71 | 0.85 | 88.72 | 43.6 | 11.43 | **61** |
| YOLOv5l | 94.25 | 89.63 | 0.92 | 94.39 | 179.51 | 47.06 | 28 |
| MBAN | **94.94** | 90.92 | **0.93** | **96.55** | 175.46 | 46 | 28 |

Note:
The experimental results of different algorithms on the NWPU VHR-10 dataset. The best results are highlighted in bold.

**Table 6  The AP (%) in each category of the NWPU VHR-10 dataset.**

| Category | SSD | YOLOv3 | EfficientNet-YOLOv3 | YOLOv4 | GhostNet-YOLOv4 | YOLOv5l | MBAN |
|---|---|---|---|---|---|---|---|
| Airplane | 99.98 | **100** | 99.79 | **100** | 99.98 | **100** | **100** |
| Ship | 81.32 | 88.12 | 91.67 | 91.73 | 84.89 | 93.84 | **95.06** |
| St | 99.29 | 98.35 | 99.44 | 98.79 | 97.91 | **99.88** | 99.80 |
| Bd | 97.27 | 95.48 | 92.42 | 95.19 | 91.71 | **99.73** | 98.65 |
| Tc | 93.28 | **99.97** | 90.89 | 98.89 | 96.97 | **99.97** | 99.84 |
| Bc | **100** | **100** | 98.89 | **100** | **100** | **100** | **100** |
| Gtf | **100** | **100** | **100** | **100** | **100** | **100** | **100** |
| Harbor | 85.40 | 91.28 | 86.68 | 90.18 | 78.10 | **100** | **100** |
| Bridge | 56.68 | 44.44 | 55.34 | 52.17 | 54.95 | 53.82 | **74.93** |
| Vehicle | 89.67 | 94.36 | 79.84 | **98.42** | 82.69 | 96.66 | 97.23 |

Note:
The AP (%) in each category of the NWPU VHR-10 dataset. The best results are highlighted in bold.

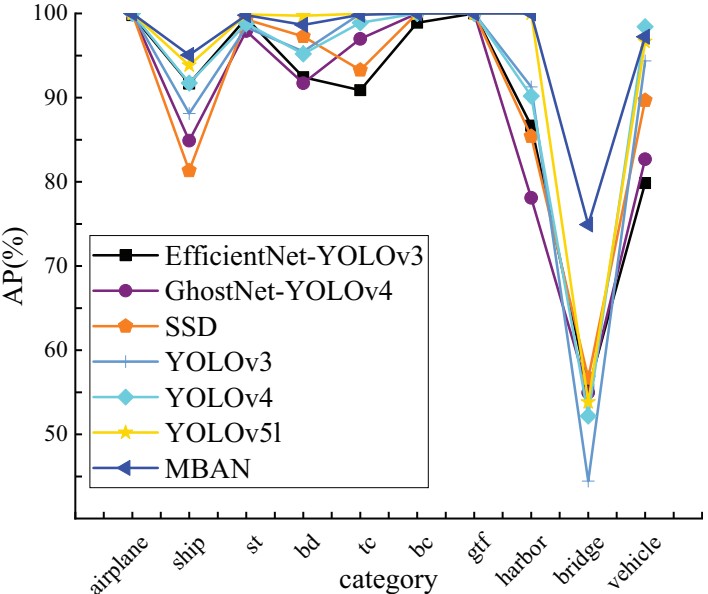

**Figure 4  The comparison of AP of 10 categories on the NWPU VHR-10 dataset.**

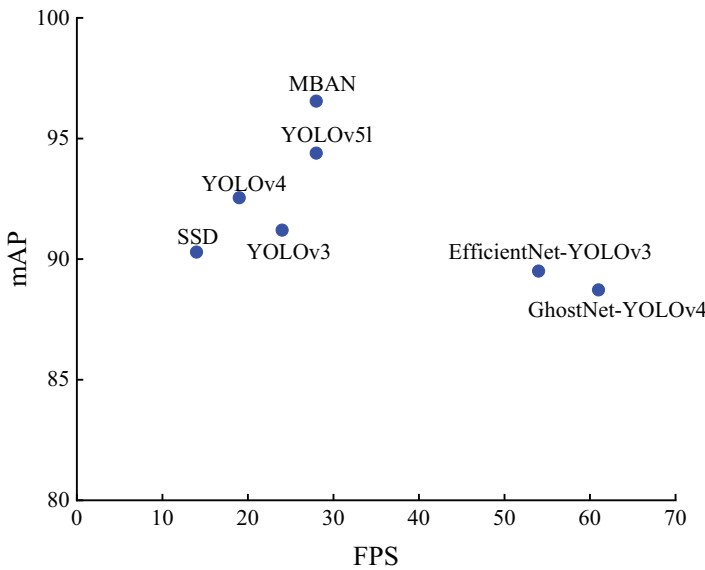

**Figure 5 The distribution of accuracy and speed with different algorithms on the NWPU VHR-10 dataset.**

respectively. The number of parameters in MBAN has decreased by 2.3%, allowing it to maintain the same detection speed while improving detection accuracy. Note: The ten categories of NWPU VHR-10 are airplane, ship, storage tank, baseball diamond, tennis court, basketball court, ground track field, harbor, bridge and vehicle, among them, storage tank, baseball diamond, tennis court, basketball court, ground track field, use st (storage tank), bd (baseball diamond), tc (tennis court), bc (basketball court), gtf (ground track field) to refer to these categories respectively.

Figure 4 visually presents the results obtained from Table 6, clearly indicating that MBAN consistently outperforms in the majority of categories.

Figure 5 shows the distribution of detection accuracy and detection speed of different algorithms on the NWPU VHR-10 dataset. It can be seen that MBAN outperforms SSD, YOLOv3 and YOLOv4 in terms of speed and outperforms SSD, YOLOv3, YOLOv4, EfficientNet-YOLOv3, GhostNet-YOLOv4 and YOLOv5l in terms of mAP. On the whole, we proposed MBAN achieves better detection performance on NWPU VHR-10 dataset.

## Discussion of results on PASCAL VOC dataset

Table 7 presents the detection results of the proposed MBAN on the PASCAL VOC dataset, in comparison our network to the YOLOv3, YOLOv4, SSD, EfficientNet-YOLOv3, GhostNet-YOLOv4, and YOLOv5l. The best results are highlighted in bold.

From Table 7, it is clear that the mAP of MBAN achieves 84.96%, and the detection speed remains at 28 FPS. In comparison to YOLOv5l, under the premise that the size of the network has decreased by 4.05 MB, and the amount of parameters is reduced by 2.3%, the $R$, $P$, $F1$ and the mAP are improved by 0.79%, 0.96%, 0.01, and 1.46 respectively, which validates that the proposed network is capable of significantly increasing detection precision while meeting the requirements of real-time detection. Compared with SSD,

**Table 7 The experimental results of different algorithms on the PASCAL VOC dataset.**

| Algorithm | R/% | P/% | F1 | mAP/% | Size/MB | Params/M | FPS |
|---|---|---|---|---|---|---|---|
| SSD | 18.19 | 77.62 | 0.27 | 41.12 | 100.27 | 26.29 | 14 |
| YOLOv3 | 64.61 | 85.42 | 0.71 | 77.86 | 236.32 | 61.95 | 24 |
| EfficientNet-YOLOv3 | 64.67 | 83.48 | 0.72 | 75.98 | **41.11** | **7.22** | 55 |
| YOLOv4 | **75.69** | 86.81 | **0.80** | 84.73 | 245.53 | 64.36 | 19 |
| GhostNet-YOLOv4 | 65.11 | 86.97 | 0.74 | 78.22 | 43.6 | 11.43 | **57** |
| YOLOv5l | 70.07 | 89.09 | 0.78 | 83.50 | 179.51 | 47.06 | 28 |
| MBAN | 70.86 | **90.05** | 0.79 | **84.96** | 175.46 | 46 | 28 |

Note:
The experimental results of different algorithms on the PASCAL VOC dataset. The best results are highlighted in bold.

MBAN has improved mAP by 43.84% and detection speed by 14 FPS. Compared with YOLOv3, MBAN has increased mAP by 7.1% and detection speed by 4 FPS. Compared with EfficientNet-YOLOv3, although the detection speed has decreased by 27 FPS, the mAP of MBAN has increased by 8.98%. Compared with YOLOv4, MBAN has increased mAP by 0.23% and detection speed by 9 FPS. Compared with GhostNet-YOLOv4, although the detection speed has decreased by 29 FPS, the mAP of MBAN has increased by 6.74%, which can still meet real-time detection requirements.

In order to analyze the detection performance of MBAN, this article also compares the detection accuracies of MBAN and some mainstream algorithms for each category on the PASCAL VOC dataset, as shown in Table 8. The best results are highlighted in bold. It is evident that MBAN outperforms the mainstream algorithms in terms of accuracy in most of the categories, and obtains the optimal detection results on nine object categories. In comparison to YOLOv5l, MBAN optimizes the YOLOv5l to more accurately capture the features, and achieves higher detection accuracy on 16 categories, and for the small objects categories, for instance, aero, bus, chair, sheep, and plant, the detection accuracies are improved by 1.8%, 2.74%, 2.73%,1.37% and 10.86%, respectively, while the detection accuracy on 4 categories lower compared to YOLOv5l, MBAN not only maintains the same detection speed, but also decreases the number of parameters and enhances the mAP by 1.46%, which gives a better performance on most categories. Compared with SSD, MBAN has improved detection accuracy in all 20 categories. Compared with YOLOv3, MBAN has improved detection accuracy in 18 categories, with only two categories having lower detection accuracy than YOLOv3. Compared with EfficientNet-YOLOv3, MBAN has improved detection accuracy in 19 categories, with lower detection accuracy in one category. Compared with YOLOv4, MBAN has improved detection accuracy in 10 categories. Although the detection accuracy in the other 10 categories is lower than YOLOv4, the YOLOv4 algorithm has a large number of parameters, and its detection speed and mAP are lower than MBAN. Compared with GhostNet-YOLOv4, MBAN has improved detection accuracy in all 20 categories. Since YOLOv3, EfficientNet-YOLOv3, YOLOv4, and YOLOv5l use different algorithm structures, they are made to have better feature learning ability on individual categories, thus achieving better detection on individual categories.

**Table 8 The AP (%) in each category of the PASCAL VOC dataset.**

| Category | SSD | YOLOv3 | EfficientNet-YOLOv3 | YOLOv4 | GhostNet-YOLOv4 | YOLOv5l | MBAN |
|---|---|---|---|---|---|---|---|
| Aero | 57.26 | 88.57 | 84.34 | **94.66** | 87.15 | 91.76 | 93.56 |
| Bike | 50.12 | 89.23 | 86.48 | 92.00 | 84.92 | 93.24 | **94.00** |
| Bird | 27.98 | 82.96 | 79.85 | 85.62 | 78.23 | 85.15 | **86.26** |
| Boat | 26.59 | 61.36 | 65.53 | 70.47 | 70.92 | 76.15 | **76.98** |
| Bottle | 4.02 | 71.06 | 49.89 | **78.52** | 63.72 | 75.49 | 75.37 |
| Bus | 57.69 | 90.35 | 81.99 | 92.83 | 80.78 | 90.75 | **93.49** |
| Car | 71.61 | 93.13 | 90.15 | 94.80 | 89.75 | 94.84 | **94.92** |
| Cat | 51.02 | 86.54 | 82.85 | 86.48 | 87.17 | **88.76** | 88.22 |
| Chair | 16.30 | 65.28 | 51.60 | 67.59 | 59.24 | 70.24 | **72.97** |
| Cow | 40.08 | 82.63 | 84.47 | **92.83** | 82.65 | 90.14 | 91.25 |
| Table | 20.37 | 76.84 | 58.94 | **80.12** | 73.52 | 74.05 | 75.03 |
| Dog | 41.26 | 81.77 | 84.77 | **91.77** | 84.55 | 86.26 | 87.78 |
| Horse | 52.50 | 90.37 | 89.71 | **95.15** | 89.80 | 92.07 | 92.16 |
| Mbike | 54.74 | 89.05 | 84.94 | **93.14** | 82.40 | 91.88 | 93.01 |
| Person | 66.14 | 89.04 | 83.41 | 91.33 | 85.17 | **91.36** | 91.27 |
| Plant | 0.97 | 52.20 | 42.38 | 55.62 | 47.87 | 44.97 | **55.83** |
| Sheep | 42.58 | 17.68 | 80.75 | **85.40** | 83.04 | 82.62 | 83.99 |
| Sofa | 38.42 | 80.41 | 71.12 | 77.05 | 73.35 | 78.09 | **82.29** |
| Train | 50.64 | 86.99 | 87.28 | 87.09 | 86.08 | **88.00** | 86.61 |
| TV | 52.11 | 81.76 | 79.05 | 82.10 | 74.19 | 84.15 | **84.31** |

Note:
The AP (%) in each category of the PASCAL VOC dataset. The best results are highlighted in bold.

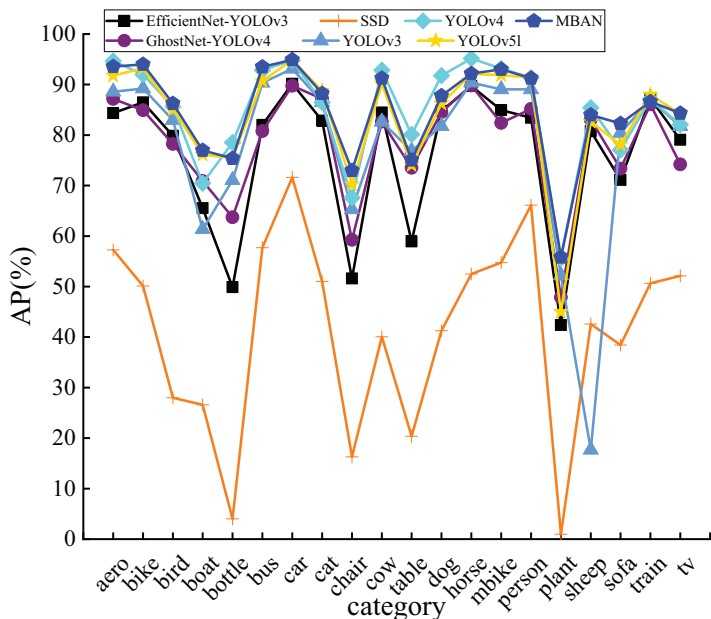

**Figure 6 The comparison of AP of 20 categories on the PASCAL VOC dataset.**

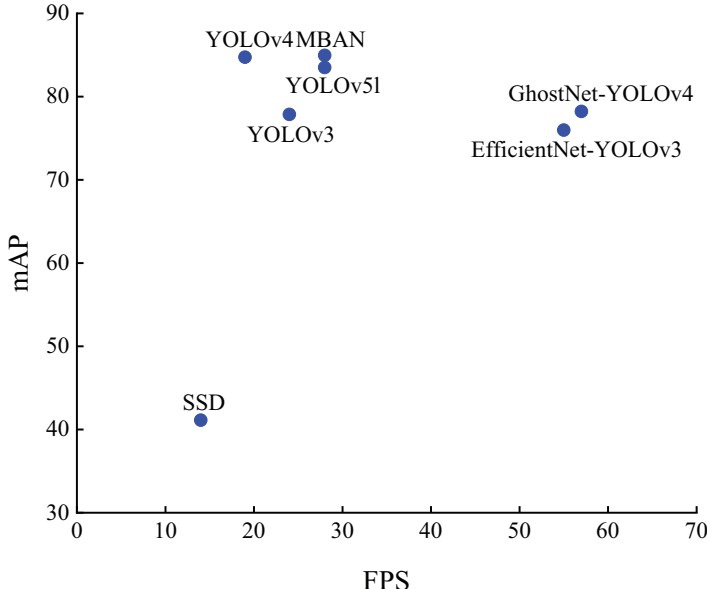

**Figure 7** **The distribution of accuracy and speed with different algorithms on the PASCAL VOC dataset.**

**Table 9 The results of ablation experiments on the NWPU VHR-10 dataset.**

| Group | MBAM | ACR | mAP (%) | FPS |
|---|---|---|---|---|
| Experiment A | × | × | 94.39 | 28 |
| Experiment B | × | √ | 95.75 | 28 |
| Experiment C | √ | × | 96.30 | 28 |
| Experiment D | √ | √ | **96.55** | 28 |

**Note:**
The results of ablation experiments on the NWPU VHR-10 dataset. The best result is highlighted in bold.

The results in Fig. 6 confirm that MBAN performs better than other algorithms. Moreover, it is apparent that MBAN has reached the highest point in majority of the categories, which also indicates that MBAN has outstanding ability in small object detection.

Figure 7 shows the distribution results on the PASCAL VOC dataset. It can be seen that MBAN outperforms SSD, YOLOv3 and YOLOv4 in speed and outperforms SSD, YOLOv3, YOLOv4, EfficientNet-YOLOv3, GhostNet-YOLOv4 and YOLOv5l in mAP. In conclusion, the MBAN proposed achieves remarkable detection performance in the aspect of detection accuracy while keeping the detection speed constant. The best results are highlighted in bold.

## Ablation experiments
A comprehensive evaluation of the proposed MBAN was conducted to assess the efficacy of each method on the NWPU VHR-10 datasets, respectively. The best result is highlighted in bold. Experiment A is the YOLOv5l algorithm, Experiment B uses the ACR, Experiment C uses the MBAM, and Experiment D combines the methods of Experiment B and

**Table 10 The results of ablation experiments on the PASCAL VOC dataset.**

| Group | MBAM | ACR | mAP (%) | FPS |
|---|---|---|---|---|
| Experiment A | × | × | 83.50 | 28 |
| Experiment B | × | √ | 83.94 | 28 |
| Experiment C | √ | × | 83.95 | 28 |
| Experiment D | √ | √ | **84.96** | 28 |

**Note:**
The results of ablation experiments on the PASCAL VOC dataset. The best result is highlighted in bold.

Experiment C. Table 9 presents the results of the ablation experiments conducted on the NWPU VHR-10 dataset, where "√" denotes the utilization of the proposed method and "×" indicates its absence.

Experiment A denotes the YOLOv5l object detection algorithm with mAP of 94.39% and FPS of 28. Experiment B uses the ACR, and the mAP improves from 94.39% to 95.75% and the FPS is 28 compared to Experiment A, which proves that the ACR can more accurately locate the position of objects. Experiment C uses the MBAM, and the mAP increases from 94.39% to 96.30% compared to Experiment A. The FPS is still 28, which proves that the effectiveness of MBAM. Experiment D combines the MBAM and the ACR, which greatly improves the detection performance, the FPS is still 28, and the mAP is improved from 94.39% to 96.55% compared with experiment A. Notably, it can be concluded that according to the design of the two methods can be employed to strengthen the detection accuracy of the network efficiently without change in detection speed, to fulfill the demands for real-time detection.

Table 10 displays the results of the ablation experiments conducted on the PASCAL VOC dataset. The best result is highlighted in bold.

Experiment A is the YOLOv5l object detection algorithm with mAP of 83.50% and FPS of 28. Experiment B uses the ACR, and the mAP is improved from 83.50% to 83.94% compared with Experiment A, which proves that the ACR can more accurately locate the position of objects, thereby improving the object detection accuracy. Experiment C uses the MBAM and the mAP is improved from 83.50% to 83.95% compared to Experiment A. The FPS is still 28, which proves that the MBAM can raise the detection accuracy of the algorithm well. Experiment D combines the MBAM and the ACR, which greatly improves the detection performance, the FPS is still 28, and the mAP is improved from 83.50% to 84.96% compared with experiment A. The results indicate that the integration of the two methods can be employed to strengthen the detection accuracy of the algorithms efficiently.

## CONCLUSION

This article proposes a novel MBAN. In order to make the network focus on the salient feature information of small objects and reduce downsampling information loss, an MBAM is designed to enhance the feature expression ability of small objects. Besides, we use the ACR method to cluster small object datasets and improve the regression and localization accuracy of small objects. The experimental results on the NWPU VHR-10

and PASCAL VOC datasets show that MBAN outperforms most popular algorithms in detection performance, achieving superior detection performance without sacrificing detection speed. Although the network proposed in this article has significant advantages in small object detection, it still needs improvement in future work for small objects with severe occlusion.

## ACKNOWLEDGEMENTS

The authors extend their thanks to the PyTorch developers. The authors would like to thank the reviewers and editors for thoughtful comments and valuable suggestions.

### Funding

The work was supported by Science and Technology Research and Development Plan Project of Handan, Hebei Province, China (21422031289) and the Ministry of Education University-Industry Collaborative Education Program, China (220601828023121). The funders had no role in study design, data collection and analysis, decision to publish, or preparation of the manuscript.

### Grant Disclosures

The following grant information was disclosed by the authors:
Science and Technology Research and Development Plan Project of Handan:
21422031289.
Ministry of Education University-Industry Collaborative Education Program:
220601828023121.

### Competing Interests

The authors declare that they have no competing interests.

### Author Contributions

- Li Li conceived and designed the experiments, performed the experiments, analyzed the data, performed the computation work, prepared figures and/or tables, authored or reviewed drafts of the article, and approved the final draft.
- Shuaikun Gao conceived and designed the experiments, performed the experiments, analyzed the data, performed the computation work, prepared figures and/or tables, authored or reviewed drafts of the article, and approved the final draft.
- Fangfang Wu analyzed the data, performed the computation work, prepared figures and/or tables, authored or reviewed drafts of the article, and approved the final draft.
- Xin An conceived and designed the experiments, performed the experiments, analyzed the data, performed the computation work, prepared figures and/or tables, authored or reviewed drafts of the article, and approved the final draft.

### Data Availability

The code is available at GitHub and Zenodo:

- https://GitHub.com/Gaoshuaikun/MBAN.

- Gsk. (2023). Gaoshuaikun/MBAN: v1.0.2 (v1.0.2). Zenodo. https://doi.org/10.5281/zenodo.10393904.

The data is available at:

- PASCAL VOC dataset: http://host.robots.ox.ac.uk/pascal/VOC/.

- PASCAL VOC 2007 dataset: http://host.robots.ox.ac.uk/pascal/VOC/voc2007/index.html.

- PASCAL VOC 2012 dataset: http://host.robots.ox.ac.uk/pascal/VOC/voc2012/index.html.

- NWPU VHR-10: https://GitHub.com/Gaoshuaikun/NWPU-VHR-10.

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
