# Peer review of "MBAN: multi-branch attention network for small object detection"

_PeerJ Computer Science, doi:10.7717/peerj-cs.1965_

## Round 0.1 · original submission · Major Revisions

The authors should revise the article to improve technical writing, explaination of equations, and research contributions.

**Language Note:** The review process has identified that the English language must be improved. PeerJ can provide language editing services - please contact us at copyediting@peerj.com for pricing (be sure to provide your manuscript number and title). Alternatively, you should make your own arrangements to improve the language quality and provide details in your response letter. – PeerJ Staff

Reviewer 1 ·

Basic reporting

1. Issues in the abstract: The abstract has several weaknesses, including a lack of clarity and precision in its language, insufficient details on specific challenges in small object detection, and a failure to provide concise explanations for key components of the proposed algorithm, such as the Multi-branch Attention Module (MBAM) and the SimAM attention mechanism. Additionally, the abstract lacks a comparative analysis of how MBAN addresses identified challenges compared to existing methods, and it doesn't offer sufficient information on the experimental methodology or validation metrics used.

2. The motivation for this research is missing. Authors should add a few lines about the motivation just before the contributions.

3. I suggest that the authors proofread the entire paper and improve its quality of writing.

Experimental design

1. The discussion of the mathematical equations is very limited.

2. The author does not declare the structure of their proposed algorithm. I suggest that the authors provide an algorithmic representation of the proposed algorithm.

3. Authors are required to add the utilized hyperparameters, in the form of a table, for the implementation of the method.

Validity of the findings

The discussion of the results especially of the table 3 - 6 is very limited.

Authors are required to justify achieving such high performance as depicted in these tables.

The conclusion section can be improved by emphasizing the most important findings (numerical results), highlighting the limitations of the research, and making recommendations for future research.

Cite this review as

Reviewer 2 ·

Basic reporting

1. The authors have presented a novel multi-branch attention module that consists of two designs; a multi-branch structure and the SimAM parameter-free attention mechanism.
2. The concept presented in this work is interesting, however, the authors can combine Contributions 1 & 2 to make a single contribution in the Introduction section.
3. The literature should discuss multi-branch attention methods used previously.

Experimental design

Which YOLO network is selected for the integration of novel MBAN module?

Validity of the findings

no comment

Additional comments

Line 10 --> has
Line 13 --> object
Line 21 --> demonstrates
Line 36 --> uses
Line 57 --> an MBAN
Line 75 --> outlined
Line 149 --> a strengthened
Line 189 --> an object/objects
Line 204 --> makes
Line 389 --> Remove one “be”

Cite this review as

---

## Round 0.2 · accepted · Accept

The authors have revised the article considerably and reviewers agree on acceptance.

Reviewer 1 ·

Basic reporting

All issues are addressed.

Experimental design

All issues are addressed.

Validity of the findings

All issues are addressed.

Cite this review as

Reviewer 2 ·

Basic reporting

Issues have been resolved already.

Experimental design

no comment

Validity of the findings

no comment

Additional comments

no comment

Cite this review as